# Effects of Teosinte Flour (*Dioon mejiae*) on Selected Physicochemical Characteristics and Consumer Perceptions of Gluten-Free Cocoa Cookies Formulated with Mung Bean (*Vigna radiata*) Flour

**DOI:** 10.3390/foods13060910

**Published:** 2024-03-17

**Authors:** Carlos José Rivera, Ricardo S. Aleman, Jorge Ortega, Andrea Muela, Jhunior Marcia, Joan King, Witoon Prinyawiwatkul

**Affiliations:** 1Faculty of Technological Sciences, Universidad Nacional de Agricultura, Catacamas 16201, Hondurasjmarcia@unag.edu.hn (J.M.); 2School of Nutrition and Food Sciences, Louisiana State University, Agricultural Center, Baton Rouge, LA 70803, USAjking@agcenter.lsu.edu (J.K.)

**Keywords:** cookie, teosinte, mung bean, gluten-free, physicochemical characteristics, consumer perception

## Abstract

*Dioon mejiae*, or teosinte, is a living fossil tree discovered in Olancho, Honduras, whose seeds have a desirable nutritional profile that can provide health benefits. As a result, the objective of this study was to evaluate the effects of teosinte flour obtained from seeds on selected physicochemical characteristics and consumer perceptions of gluten-free cocoa cookies formulated with mung bean (*Vigna radiata*) flour. Gluten-free cocoa cookies were prepared with different levels of teosinte flour (0%, 70%, 80%, 90%, and 100% by weight of mung bean flour) in substitution of mung bean flour. The cookies were evaluated for texture hardness, color (L*, a*, b*), moisture content, and water activity. Sensory acceptability of appearance, color, texture, aroma, flavor, and overall quality of cocoa cookies was rated by 175 consumers using a “yes/no” binomial scale. Overall liking was evaluated using a 9-point hedonic scale. Purchase intent was evaluated with a “yes/no” binomial scale. The levels of teosinte flour did not significantly affect the acceptability of appearance, color, texture, flavor, aroma, and overall quality, and neither the overall liking nor the purchase intent responses. However, the texture attribute had the lowest % acceptability response among all sensory attributes. The addition of teosinte flour did not affect water activity and color (L*, a*, b*), whereas it decreased the texture hardness (g force), producing softer cookies. Cocoa cookies stayed acceptable even at 100% teosinte flour addition (70% acceptability; mean overall liking = 5.69). Teosinte flour has an excellent nutritional profile that could be practically applied in baked goods.

## 1. Introduction

Due to their characteristics and a high carbohydrate content, cookies are foods with a more complex and crunchy texture when compared to products low in carbohydrate content [1]. This product is prevalent worldwide; the market is increasingly growing with new varieties or formulas adapted from the original product by replacing ingredients to add value or to make it more attractive to consumers [2,3]. However, excessive consumption of energy-dense cookies can lead to adverse effects on health. Intake beyond recommended levels may lead to elevated triglyceride levels, increased insulin production, obesity, reduced appetite, hyperactivity, and an increased risk of cardiovascular diseases [4]. To address this concern, exploration of alternative ingredients becomes paramount. Most cookies commercially available are elaborated from wheat flour [5]. Teosinte and mung bean (*Vigna radiata*) are emerging as excellent substitutes for conventional wheat flours in cookie formulations. These ingredients provide unique nutritional profiles and potential health benefits, making them promising options for creating more nutritious and wholesome gluten-free cookies. These alternative ingredients offer the potential to increase protein content and beneficial dietary fiber and provide a range of essential vitamins and minerals. The inclusion of such ingredients aligns with the growing demand for healthier gluten-free snack options and can contribute to diversifying the cookie market with innovative and nutritionally enriched offerings [6,7,8].

Teosinte holds a significant cultural and historical importance to Honduras, where it has been utilized since ancient times and is widely distributed in the northeastern regions [6]. This cycad plant serves a dual purpose in the region, serving as a vital food source for over 30,000 Hondurans and playing a role in ceremonial ornamentation across numerous communities throughout the country [7]. Cycads, including teosinte, belong to a unique group of plants that emerged long before the division between monocots and dicotyledons. It is important to note that teosinte, despite sharing a common name, is not related to the subspecies of the ancestor of *Zea mays*, commonly referred to as teosinte, teosintle or tiusinte [6]. The distinction between these plants is crucial when discussing their botanical classification and evolutionary history. The utilization of teosinte as a food source highlights the deep-rooted connection between plants and human culture, where ancient communities recognized and harnessed the nutritional benefits of this plant. Likewise, the study published by Bastias et al. (2020) [6] found that teosinte flour is a caloric food containing mainly starch. The protein content (9.67 g/100 g of flour) was similar to that of other cereals. This cycad flour could provide amino acids, including leucine, glutamic acid, proline, tyrosine, phenylalanine, and arginine, leucine, and especially lysine, at a level of more than 25% of most amino acids for an adult of 70 kg. In addition, teosinte flour had a high content of unsaturated fatty acids, predominantly oleic acid (C18:1) and linoleic acid (C18:2) [6]. Regarding minerals, teosinte flour presents Fe (6.85 mg/100 g of flour), Zn (1.46 mg/100 g of flour), Ca (14.86 mg/100 g of flour) and P (241 mg/100 g of flour) [6]. So, teosinte flour has nutrients and qualities that give it excellent nutritional and beneficial capacities for health, as well as it being a potential ingredient to be used mainly in mixtures with other flours.

On the other hand, mung bean is a valuable food source with an important significance, especially in Asian cuisines. It is consumed in various forms, such as a side dish, a dessert, bread and even boiled or cooked mixed with vegetables and meat [8]. Mung beans have arisen as a subject of increasing scientific inquisitiveness due to their potential health-promoting characteristics [9]. Mung beans have been frequently described as being rich in protein, micronutrients, and polyphenols. This type of bean has various nutrients that give it its therapeutic power, such as bioactive compounds, proteins, minerals, and fiber. It has been proven that it helps reduce blood glucose, cholesterol, and triglycerides, has antihypertensive effects, protects the liver, and even helps prevent cancer. Mung beans can be consumed together with foods from the cereal group, which helps provide better-quality protein. These compounds have been shown to have anti-diabetic, antimicrobial, antihypertensive, antioxidant, and anti-hyperlipidemic properties [10]. The combination of teosinte flour’s unique nutritional qualities and the health-promoting properties of mung beans presents an opportunity for exploring synergistic effects when incorporating both ingredients into food formulations. Such an innovative blend has the potential to deliver enhanced nutritional benefits and contribute to the development of functional foods that cater to the consumers’ growing demand for health-conscious and nutritionally balanced products.

Legumes are not only an important source of food, but they also play a crucial role in addressing protein deficiencies, particularly in developing countries [11]. According to Onwurafor et al. (2017) [12], among legumes, mung bean stands out due to its high digestibility and the absence of flatulence, making it an ideal ingredient for value-added products for infants, recuperating patients and aged people. Ahmad et al. (2021) [13] studied the effect of biscuits made with mung beans (*V. radiata*), and star gooseberry (*Sauropus androgynous*) leaves on infant weight. The study revealed that these biscuits contributed to improved weight gain in infants. On the other hand, the flour of teosinte was evaluated and did not show acute toxicity in the tested animals. This study was carried out under a limited test scheme, considering other non-acute toxicology tests (carcinogenicity, teratogenesis, and chronic toxicity). The results of the study suggested a free radical scavenging activity of the teosinte flour comparable to the standard used (Trolox) at a concentration of 25.00 mg/mL [14]. These findings hint at the potential antioxidant properties of teosinte flour, opening opportunities for further research in the field of natural antioxidants and their health benefits.

Evaluating various affective sensory modalities not only enhances our understanding of sensory techniques but also opens new possibilities for incorporating novel ingredients [15]. A crucial aspect of this evaluation involves determining the factors that influence the acceptance or rejection of a particular food item [16]. To date, no research has been conducted on the development of cocoa cookies prepared using a combination of teosinte and mung bean flours. Consequently, the primary objective of this study was to evaluate physicochemical properties and consumer perceptions of gluten-free cocoa cookies formulated with mung bean (*Vigna radiata*) flour that was substituted with teosinte (*Dioon mejiae*) flours at varying levels from 70% to 100% by weight. By incorporating teosinte flour into the cookie formulation, we aimed to assess its impact on the texture, taste, and overall acceptability of the final product. Additionally, sensory evaluations were conducted to gather valuable insights into consumer preferences that led to a consumer “rejection threshold” of teosinte flour addition. Understanding how teosinte flour interacts with mung bean flour in the context of cocoa cookies will contribute to expanding the repertoire of gluten-free alternatives and promoting the use of underutilized ingredients, i.e., teosinte flour.

## 2. Materials and Methods

### 2.1. Raw Materials

The teosinte seed coat was removed and nixtamalized with calcium hydroxide at 80 °C for 3 h. After the nixtamalization, the seed endosperm was dehydrated in a convection oven (Digitronic TFT- Selecta, J.P. SELECTA, Barcelona, Spain) at 50 °C/48 h and was ground in a knife mill Retsch SM 100 (Retsch GM200, Retsch GmbH, Haan, Germany) to obtain flour with particle size between 501 and 700 mm. Mung beans (Jiva organics, Burnaby, British Columbia, Canada) were ground for 30 s in a commercial electric grain grinder (CGOLDENWALL Inc., Hangzhou, China). Both flours were stored in sealed plastic bags at room temperature (20 °C with 70% relative humidity) until further use.

### 2.2. Preparation of Cocoa Cookies

Teosinte flour (Obtained as described in Section 2.1), mung bean flour, baking powder (Great Value^TM^, Bentonville, AR, USA), cacao powder (Better Body Foods, Lindon, UT, USA), unsalted butter (Great Value^TM^), sugar (Great Value^TM^), Splenda (Great Value^TM^), and eggs (Great Value^TM^) were mixed in a commercial food mixer (Globe Food Equipment CO, model SP5, Dayton OH, USA). The ingredients were used in the following percentages, by weight of cookie dough: flour, 42%; unsalted butter, 16%; baking soda, 2%; egg, 7%; Splenda, 6%; sucrose, 17%; and cacao powder, 10%. Mung bean flour served as the control group at 0% (MB-TF 0%) and teosinte flour was used as a substitute for mung bean flour at 70% (MB-TF 70%), 80% (MB-TF 80%), 90% (MB-TF 90%), and 100% (MB-TF 100%) to prepare the cookies (Figure 1). Doughs were rested for 5 min, and a hand roller was used to flatten and cut them into circles using a cookie cutter with a diameter of 4.83 cm. Doughs were oven baked (Baxter OV310G, Orting, WA, USA) at 345 °F for 9 min, and the cookies were then removed and cooled to room temperature. Samples were stored overnight in Ziploc^®^bags (SC Johnson, Racine, WS, USA) at room temperature prior to testing the following day.

### 2.3. Proximal Analysis of Cocoa Cookies

The proximate analysis of cocoa cookies was performed in triplicate at the LSU Agricultural Chemistry at Louisiana State University, including moisture (AACC method 44_01.01), protein (AACC Method 46_13.01), crude fiber (differential method), and fat (AACC Method 30_20.01). Moisture content was measured from the remains after drying (VWR International oven, Model 1370 GM, Sheldon Manufacturing Inc., Cornelius, OR, USA) the cookies (1 g) in aluminum pans at 105 °C for 24 h. The protein content was measured using the micro-Kjeldahl (Labconco Kansas City, MO, USA) digestor. The total dietary fiber content of the cookies was measured by AACC enzyme gravimetric analysis. The crude fat content in the cookies was measured using the Soxtec equipment (Soxtec System HT6, Tecator AB, Höganäs, Sweden) utilizing hexane as a solvent (155 to 210 °C) [17,18].

### 2.4. Physicochemical Characteristics of Cocoa Cookies

Texture hardness (g force) of the cookies was determined in ten replications using a Texture Analyzer (TA.XTplus^®^, Stable Micro Systems, Godalming, UK) through a shearing knife blade under the following test configurations: pre-test speed = 1.6 mm/s; trigger force = 30 g; test speed = 2.20 mm/s; and post-test speed of 12 mm/s with a distance of 10 mm [19]. The water activity (a_w_) of cocoa cookies was determined in triplicate using an a_w_ meter (Hygrolab, Rotronic, Hauppauge, NY, USA). The color of cocoa cookies was measured in seven replications using a colorimeter (BC-10 Baking Contrast Meter, Konica Minolta Sensing Americas, Ramsey, NJ, USA), reporting the L*, a*, b*, and Delta-E values. The moisture content of cocoa cookies was determined in triplicate utilizing a moisture analyzer (HE53, Mettler Toledo, Columbus, OH, USA) [20].

### 2.5. Enumeration of E. coli/Coliforms and Aerobic Plate Counts

Serial dilutions of cookies were prepared for microbial examinations in Phosphate Buffered Saline (PBS) solution and plated in duplicate on 3M™ petrifilms (3M Microbiology, St. Paul, MN, USA). The 3M™ *E. coli*/Coliforms and Aerobic Plate petrifilm plates were aerobically incubated for 48 h at 35 °C (AOAC Official Method 991.14 and 990.12) [21,22]. A Quebec Darkfield Colony Counter (Leica Inc., Buffalo, NY, USA) was utilized to count the colony-forming units (CFU).

### 2.6. Consumer Test of Cocoa Cookies

A total of N = 175 consumers were recruited from the Louisiana State University campus (LSU, Baton Rouge, LA, USA) to participate in the study. Prior to their involvement, participants were provided with a consent form to read and sign. To avoid any potential allergic reactions, consumers were informed that samples may contain allergens. The consumer sample consisted of 62.1% males and 37.9% females, representing a diverse range of individuals. Using the categorizations outlined by Dimock (2018) [23], the distribution of participants in terms of generational cohorts was as follows: 5.2% as boomers (age 60 to 77), 2.3% as generation X (age 44 to 59), 21.8% as millennials (generation Y; age 27 to 43), and a majority of 70.7% were identified as generation Z (age 19 to 26); the participants must be at least 18 years of age to participate in this study. It is worth noting that the prevalence of younger adults within the sample was expected, given the recruitment process taking place on a college campus. Consequently, it is important to acknowledge that the results of this study may not be generalizable for the broader population. The panelists who participated in this research represented a diverse range of racial and ethnic groups. Among the panelists, the majority identified as White/Caucasian, comprising 48.9%, the Latino community accounted for 25.3% of the panelists, while African-Americans constituted 13.8% of the group. Asian panelists represented 10.4% of the participants. Additionally, a small percentage, 1.6%, identified as belonging to multiple races, highlighting the multicultural nature of the consumer panel. The diverse racial and ethnic composition of the panelists ensured a broad representation of consumer perspectives and preferences. This inclusivity allows for a more comprehensive understanding of how different cultural backgrounds and experiences influence the acceptance and sensory perceptions of the gluten-free teosinte and mung bean cocoa cookies. The sensory analysis was conducted in partitioned booths at the LSU Sensory Services Lab (Baton Rouge, LA, USA). To facilitate the collection of data, Qualtrics software (Qualtrics, Provo, UT, USA) was used for questionnaire presentation and response collection. This research was approved by the Louisiana State University Agricultural Center Institutional Review Board (IRB# HE18-9 and IRB# HE 18-22). Sample presentation followed a balanced incomplete block design (BIB: t = 7, k = 4, b = 7, r = 4, λ = 1) [24]. With the BIB protocol, the product of k and b (4 × 7 = 28) is equal to the product of t and r (7 × 4 = 28). Every consumer was considered to be a block, and every block received four samples (k = 4). Cookies were portioned into white trays (four cookies per tray) and labelled with three-digit randomized codes to serve to the panelists. Each consumer received 4 (k = 4) of the possible 7 (t = 7) sample combinations. Therefore, one replication of the present BIB design required seven panelists (b = 7 blocks per replication). Replicating this BIB design twenty-five times, data from 175 total consumers were obtained (7 × 25 = 175 total panelists). All sample pairs were evaluated four times (r = 4) per replication, and thus, one hundred times over twenty-five replications of the design (4 × 25 = 100 total observations per sample). Sensory acceptability of appearance, color, texture, aroma, flavor, and overall quality of cocoa cookies was analyzed using a “yes/no” binomial scale. Overall liking was evaluated using a 9-point hedonic scale. Purchase intent was evaluated with a “yes/no” binomial scale. Unsalted crackers and water were served for palate cleansing.

### 2.7. Statistical Analysis

The data were analyzed using SAS software (Copyright^©^ 2016 SAS Institute Inc., Cary, NC, USA). Univariate analyses of variance (ANOVA) with post hoc Tukey test were used to compare the means of overall liking scores (9-point hedonic scales), texture hardness, color (L*, a*, b*, and Delta-E values), moisture content, and water activity across the different treatments (C0, C70, C80, C90, and C100). A 2-tailed binomial test (n = 100; α = 0.05) was applied to determine the statistical significance of the acceptability of the different treatments (C0, C70, C80, C90, and C100). A probit regression model was used to determine the rejection tolerance threshold (RTT) and rejection range (RR) [15].

## 3. Results and Discussion

### 3.1. Proximal Analysis of Gluten-Free Cocoa Cookies

The proximal analysis of gluten-free cocoa cookies made with mung bean flour and teosinte flour is presented in Table 1. The moisture (6.14% to 7.19%), protein (8.33% to 15.80%), fat (20.73% and 22.36%), and crude fiber (2.36% and 0.73%) were observed for gluten-free cocoa cookies. These ranges were noticed within similar contents of other types such as chocolate [25]. Global product development trends favor high protein and fiber content in baked products [26]. Not surprisingly, the control cookie (100% mung bean powder had high protein and fiber contents. Mung beans are rich in antioxidants and nutrients, which may promote digestive health, weight loss, cholesterol control, and regulation of blood pressure and blood sugar levels [27]. On the other hand, the nutritional value of teosinte seed is high with protein and methionine levels higher than maize [6]. Teosinte has antioxidant properties similar to vitamin C; it is also a neuroprotector that prevents diseases such as Alzheimer’s, Parkinson’s, and senile dementia and is a favorable food for patients with celiac disease and diabetes due to its low glycemic index and high dietary fiber content, not containing gluten.

### 3.2. Physicochemical Properties of Gluten-Free Cocoa Cookies

Hardness plays a crucial role in determining the acceptability of baked goods. The texture hardness of the gluten-free cocoa cookies showed a notable impact on the teosinte flour incorporation (Table 2). The gluten-free cocoa cookies made with teosinte flour reported significantly (*p* < 0.05) lower hardness (i.e., softer) when compared to cookies made with mung bean flour. Franklin et al. (2023) [28] reported that teosinte flour increased the hardness of bread formulated with high-protein white rice and high-protein brown rice. Nanyen et al. (2016) [29] reported that the break strength of the cookies decreased (1.90 kg to 1.57 kg) as the percentage of mung beans increased.

In terms of color, there were no changes in color lightness (L*) as the TF level increased (Table 2). Among the samples containing TF, the color redness (a*) and yellowness (b*) significantly increased with increased TF levels, which reflected the increase in ΔE values. The values of ΔE ranged from 0.51 to 1.01, which were below the difference threshold of ΔE ≈ 2.3, indicating no obvious color differences for human observers [30]

Regarding water activity (aw), the incorporation of teosinte flour did not have a noticeable impact on the aw values of the cookies, ranging from 0.48 to 0.52. Generally, proteins are known to bind water and reduce aw; however, the quantities of teosinte flour used in this study did not show a significant effect on aw. The aw values remained relatively consistent across all formulations, indicating that the teosinte flour did not influence the water-binding properties in a substantial manner. Overall, changes in color (a* and b*) and texture hardness may affect sensory acceptability (yes or no) of appearance, color, and texture (Figure 2), which is further discussed below. In the decision process to accept or reject a product, the first factor the consumer considers is its appearance, which influences the product even before its actual consumption. For example, in the case of packaged foods, the packaging constitutes the first and only information the consumer initially perceives about the product. Furthermore, among the different attributes associated with appearance, such as shape, size, or color, the latter is the one that stands out above the others since it offers us critical information about the product, even influencing the flavor. Currently, the idea that consumers form a prior belief about the product’s taste only by seeing its color is gaining strength in research on the origin of the color–flavor interaction considering the role of expectations and the background experiences (cognitive aspects) [31].

### 3.3. Microbial Analysis

The gluten-free cocoa cookies were made prior to the sensory analysis for safety purposes, and the critical limits for total aerobic plate counts, total coliforms/*E. coli*, *L. monocytogenes*, and *Salmonella*, were 1000, 50, and 10 CFU mL^−1^, respectively. Its purpose is to detect and determine germ content, minimize contamination risks, and prevent outbreaks of foodborne illnesses. In the results of the analysis, no batch produced surpassed the counts of the regulated limits. Food microbiology tests are essential to guarantee the quality of food products. In short, microbiology analyses in the food industry are integral to it. They must be carried out to ensure food quality and hygiene since their fundamental purpose is to identify and restrict harmful microorganisms that can spoil food and ensure safety from foodborne diseases.

### 3.4. Liking and Purchase Intent of Gluten-Free Cocoa Cookies

Table 2 shows the overall liking scores of ≥5.11 for the gluten-free cocoa cookies. The cookie containing 100% TF had the highest score, although the overall liking scores were not significantly different across the samples. Positive purchase intent before the Beneficial Information (HBI) was given to consumers ranged from 53.5% to 64.65%; the highest was observed for the MB-TF 100% sample. The positive purchase intent after HBI was slightly higher than that before HBI; however, they were not significant (*p* > 0.05). Aleman et al. (2022) [32] also reported no differences in purchase intent before and after HBI for cricket chocolate chips containing edible cricket protein. Usually, the beneficial claims are significant for initial liking [33]. Noor Aziah et al. (2011) [34] suggested that the beany flavor in legumes flour could be a challenge to overcome when developing cookies. The aftertaste could be the main reason for rejections of the beany flavor of the legume; however, this was not observed in this study due to the masking effect of cocoa added in the formulation.

The samples’ appearance, color, aroma, texture, flavor, and overall quality did not experience a substantial drop in acceptability with increasing teosinte flour levels (Figure 2). The “yes” acceptability responses for these sensory attributes were above the critical value (60% when n = 100 at α = 0.05). In other words, no rejection tolerance threshold (RTT) was found for all samples based on appearance, color, aroma, texture, flavor, and overall quality.

### 3.5. Predicting Purchase Intent Using Logistic Regression Analysis

The teosinte addition in gluten-free cocoa cookies was analyzed by ascertaining the effect of gender, age, race, appearance, color, aroma, texture, flavor, and liking on PI after HBI (Table 3). Gender, texture, and overall liking scores were the most significant predictors for PI after HBI. Based on the logistic model, the probability of purchase is higher for female consumers who are willing to consume the product upon tasting (CI = Yes). These results suggest that marketing strategies should target consumers who match this ideal “profile,” as they are more likely to purchase cocoa cookies containing teosinte and mung bean.

Similarly, another study has shown that texture and liking are the main drivers for purchase intent regarding brown rice cookies [35]. PI is related to how gender groups make purchase judgments [36], and engagement with HBI is predominantly related to women [37]. Consumers could be more aware of pricing and safety factors than the HBI concerning purchase intent.

## 4. Conclusions

This study examined the addition of teosinte flour into gluten-free cocoa cookies to improve their nutritional profile. The levels of teosinte flour did not significantly affect the acceptability of appearance, color, texture, flavor, aroma, and overall quality, nor did they effect the overall liking and purchase intent responses. Physicochemical analyses showed that teosinte flour did not affect water activity, whereas its addition decreased the texture hardness, producing softer cookies. Finding new ways to feed the future is one priority of the food industry, which could involve industrial and technological usage of new food sources. Teosinte flour has an excellent nutritional profile that could be applied to foods, providing health benefits. This study’s finding could serve academia and the industry due to its practical application. Preference mapping techniques and conjoint analysis should be used in future research for a better understanding of consumer behavior.

## Figures and Tables

**Figure 1 foods-13-00910-f001:**
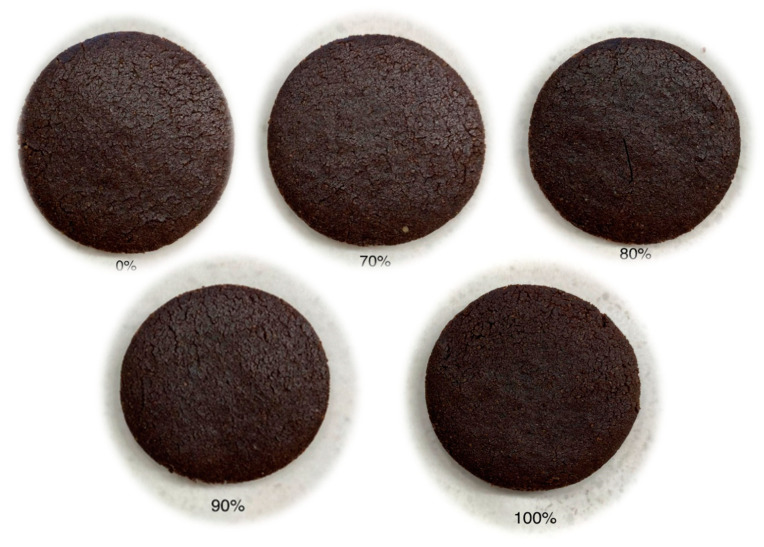
The 0%, 70%, 80%, 90%, and 100% MB-TF (mung bean–teosinte flour) treatments correspond to the TF addition at 0% (i.e., 100% MB, the control), 70%, 80%, 90%, and 100% by weight of the MB, respectively.

**Figure 2 foods-13-00910-f002:**
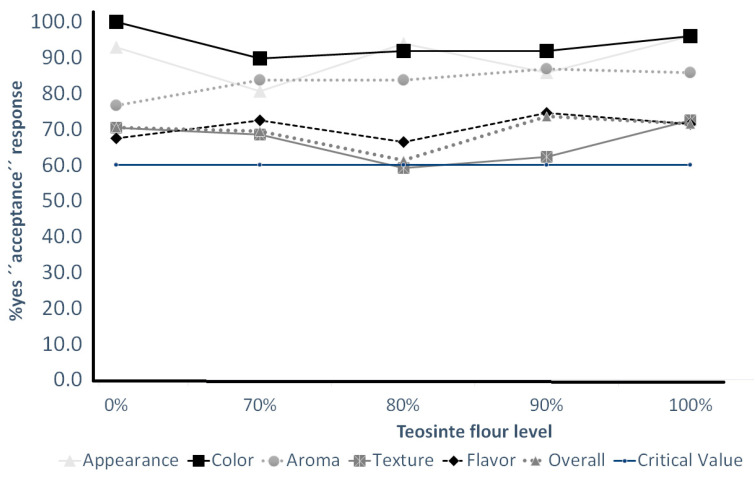
Acceptability of cocoa cookies containing teosinte flour. Percentage of ‘yes’ responses to binomial (‘yes/no’) acceptability questions for appearance, color, texture, flavor, aroma, and liking. Critical value at 60% ‘yes’ responses, based on a two-tailed binomial test for proportion (*p* = 0.50, n = 100 (data points); α = 0.05).

**Table 1 foods-13-00910-t001:** Proximate composition (%) of gluten-free cocoa cookies.

Sample *	Moisture **	Protein	Crude Fiber	Fat **
MB-TF 0%	6.68 ± 0.52 ^a^	15.8 ± 0.28 ^a^	2.36 ± 0.83 ^a^	22.36 ± 1.27 ^a^
MB-TF 70%	7.19 ± 0.27 ^a^	10.73 ± 0.11 ^b^	1.56 ± 0.40 ^ab^	21.56 ± 2.34 ^a^
MB-TF 80%	6.83 ± 1.06 ^a^	9.70 ± 0.17 ^c^	1.36 ± 0.34 ^ab^	21.36 ± 1.31 ^a^
MB-TF 90%	6.14 ± 0.14 ^a^	9.33 ± 0.11 ^c^	0.90 ± 0.15 ^b^	20.90 ± 2.01 ^a^
MB-TF 100%	6.86 ± 0.99 ^a^	8.33 ± 0.13 ^d^	0.73 ± 0.34 ^b^	20.73 ± 2.21 ^a^

^a–d^ Means followed by different letters in the column significantly differ by the Tukey’s test (*p* < 0.05). * The 0%, 70%, 80%, 90% and 100% MB-TF treatments correspond to the TF addition at 0%, 70%, 80%, 90% and 100%, respectively. ** No significant differences were detected.

**Table 2 foods-13-00910-t002:** Physicochemical properties, overall liking score and purchase intent of gluten-free cocoa cookies.

Attribute	TF Replacement Levels *				
	MB-TF 0%	MB-TF 70%	MB-TF 80%	MB-TF 90%	MB-TF 100%
L*^, NS^	27.84 ± 0.77	27.90 ± 0.95	28.51 ± 0.48	28.17 ± 0.69	27.91 ± 0.75
a*	6.77 ± 0.63 ^d^	7.03 ± 0.55 ^c^	7.32 ± 0.72 ^b^	7.48 ± 0.40 ^b^	7.65 ± 0.45 ^a^
b*	5.09 ± 0.55 ^b^	4.84 ± 0.47 ^c^	4.71 ± 031 ^c^	5.21 ± 0.35 ^a^	5.27 ± 0.39 ^a^
∆E	N/A	0.51 ± 0.12 ^b^	0.98 ± 0.26 ^a^	0.94 ± 0.36 ^a^	1.01 ± 0.37 ^a^
Water activity^NS^	0.52 ± 0.01	0.49 ± 0.03	0.48 ± 0.03	0.48 ± 0.03	0.50 ± 0.02
Hardness (g force)	12432 ± 134 ^a^	8808 ± 78 ^c^	9343 ± 189 ^b^	7692.26 ± 87 ^d^	7556 ± 94 ^d^
Overall Liking^NS^	5.11 ± 1.63	5.54 ± 1.33	5.68 ± 1.18	5.62 ± 1.55	5.69 ± 1.55
^+^Purchase intent (%)					
Before^NS^	62.00	61.62	56.12	53.54	64.65
After	69.00	64.65	62.24	56.57	70.71

^a–d^ Means followed by different letters in the row significantly differ by the Tukey test (*p* < 0.05). * The 0%, 70%, 80%, 90% and 100% MB-TF treatments correspond to the TF addition at 0%, 70%, 80%, 90% and 100%, respectively. NS = No significant differences (*p* > 0.05). + No significant differences (*p* > 0.05) were found for all treatments in purchase intent based on the McNemar’s test, comparing before and after consumers had been given the information about products health benefits.

**Table 3 foods-13-00910-t003:** The odds ratio for predicting purchase intent after beneficial statement.

Attributes		
	Pr > *X*^2^	Odds Ratio
Gender	**0.0914**	1.960
Age	0.4875	1.184
Race	0.4095	1.084
Appearance	0.2389	0.301
Color	0.7142	0.689
Aroma	0.1341	0.321
Texture	**0.0273**	0.279
Flavor	0.9266	0.928
Liking	**<0.001**	0.030

Based on logistic regression analysis, using a full model. Analysis of maximum likelihood estimates was used to obtain parameter estimates. The significance of parameter estimates was based on the Wald *X*^2^ value at *p* < 0.10.

## Data Availability

The data that support the findings of this study are available from the corresponding author upon reasonable request.

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
