# Peer review of "Effects of Teosinte Flour (Dioon mejiae) on Selected Physicochemical Characteristics and Consumer Perceptions of Gluten-Free Cocoa Cookies Formulated with Mung Bean (Vigna radiata) Flour"

_foods, 2024, doi:10.3390/foods13060910_

Round 1

Reviewer 1 Report

Comments and Suggestions for Authors

This study evaluated the influence of teosinte flour on the physicochemical attributes and consumer perceptions of gluten-free cocoa biscuits crafted from mung bean flour. In general, the findings reveal that teosinte flour boasts a commendable nutritional profile and holds promise for practical incorporation into gluten-free baked products. Nonetheless, certain inquiries persist:

(1) Title: ‘Effects Of Teosinte Flour (Dioon mejiae) on Physicochemical Characteristics And Consumer Perceptions of Gluten-Free Co-coa Cookies Formulated With Mung Bean (Vigna radiata) Flour’

In reality, the author solely assessed the texture, color, and organoleptic attributes of the biscuits, and the emphasis on Physicochemical Characteristics is excessive. Kindly revise accordingly.

(2) Abstract: When composing the abstract, strive to minimize the utilization of intricate terminology, employ acronyms judiciously, and maintain clarity and simplicity to facilitate rapid comprehension of the study's primary content and significance by readers. Furthermore, ensure that the abstract maintains logical coherence with seamless transitions between its components. Typically, the abstract length for SCI papers ranges between 150-250 words. Please review the Guide for Authors and refine the abstract accordingly to meet the specific requirements of the Foods journal.

Materials and methods:

(3) Materials and sources utilized within the text ought to be delineated in a distinct section (eg. 2.1 Materials), and redundancy of sources across subsequent sections must be circumvented.

(4) 2.1 Plan material: This section primarily outlines the method for extracting Teosinte flour rather than providing a description of the material. Furthermore, the occurrence of consecutive parentheses, such as (Retsch GmbH, Haan, Germany) (501-700 mm), etc., should be avoided.

The article exhibits some formatting issues; please address and rectify them.

Comments on the Quality of English Language

This study evaluated the influence of teosinte flour on the physicochemical attributes and consumer perceptions of gluten-free cocoa biscuits crafted from mung bean flour. In general, the findings reveal that teosinte flour boasts a commendable nutritional profile and holds promise for practical incorporation into gluten-free baked products. Nonetheless, certain inquiries persist:

(1) Title: ‘Effects Of Teosinte Flour (Dioon mejiae) on Physicochemical Characteristics And Consumer Perceptions of Gluten-Free Co-coa Cookies Formulated With Mung Bean (Vigna radiata) Flour’

In reality, the author solely assessed the texture, color, and organoleptic attributes of the biscuits, and the emphasis on Physicochemical Characteristics is excessive. Kindly revise accordingly.

(2) Abstract: When composing the abstract, strive to minimize the utilization of intricate terminology, employ acronyms judiciously, and maintain clarity and simplicity to facilitate rapid comprehension of the study's primary content and significance by readers. Furthermore, ensure that the abstract maintains logical coherence with seamless transitions between its components. Typically, the abstract length for SCI papers ranges between 150-250 words. Please review the Guide for Authors and refine the abstract accordingly to meet the specific requirements of the Foods journal.

Materials and methods:

(3) Materials and sources utilized within the text ought to be delineated in a distinct section (eg. 2.1 Materials), and redundancy of sources across subsequent sections must be circumvented.

(4) 2.1 Plan material: This section primarily outlines the method for extracting Teosinte flour rather than providing a description of the material. Furthermore, the occurrence of consecutive parentheses, such as (Retsch GmbH, Haan, Germany) (501-700 mm), etc., should be avoided.

The article exhibits some formatting issues; please address and rectify them.

Author Response

This study evaluated the influence of teosinte flour on the physicochemical attributes and consumer perceptions of gluten-free cocoa biscuits crafted from mung bean flour. In general, the findings reveal that teosinte flour boasts a commendable nutritional profile and holds promise for practical incorporation into gluten-free baked products. Nonetheless, certain inquiries persist:

(1) Title: ‘Effects Of Teosinte Flour (Dioon mejiae) on Physicochemical Characteristics And Consumer Perceptions of Gluten-Free Co-coa Cookies Formulated With Mung Bean (Vigna radiata) Flour’

In reality, the author solely assessed the texture, color, and organoleptic attributes of the biscuits, and the emphasis on Physicochemical Characteristics is excessive. Kindly revise accordingly.

Answer: Thank you for comment, the title was slightly modified.

(2) Abstract: When composing the abstract, strive to minimize the utilization of intricate terminology, employ acronyms judiciously, and maintain clarity and simplicity to facilitate rapid comprehension of the study's primary content and significance by readers. Furthermore, ensure that the abstract maintains logical coherence with seamless transitions between its components. Typically, the abstract length for SCI papers ranges between 150-250 words. Please review the Guide for Authors and refine the abstract accordingly to meet the specific requirements of the Foods journal.

Answer: Abstract was improved and kept within 250 words.

Materials and methods:

(3) Materials and sources utilized within the text ought to be delineated in a distinct section (eg. 2.1 Materials), and redundancy of sources across subsequent sections must be circumvented.

Answer: This section was modified.  Raw Materials and Preparation of Cocoa Cookies were separated into 2 sections.

(4) 2.1 Plan material: This section primarily outlines the method for extracting Teosinte flour rather than providing a description of the material. Furthermore, the occurrence of consecutive parentheses, such as (Retsch GmbH, Haan, Germany) (501-700 mm), etc., should be avoided.

Answer:  As suggested, more information was provided, and the occurrence of consecutive parentheses was avoided.

The article exhibits some formatting issues; please address and rectify them.

Answer:  Thank you.  Your recommendation was followed.

Reviewer 2 Report

Comments and Suggestions for Authors

The manuscript evaluates the effects of teosinte flour (Dioon mejiae) on physicochemical characteristics and consumer perceptions of gluten-free cocoa cookies formulated with mung bean (Vigna radiata) flour. The topic is interesting; However, the manuscript has several problems:

1. Check the spacing between words. For instance, L 42; worldwide; the, or L 43; from the... Check the entire manuscript.

2. L 51-58 needs some references.

3. Please check the reference style. For instance, L 71; "Bastias et al. [6]" not "Bastias et al. (2020) [6]".

4. L 80-86; you can elaborate more on the nutritional characteristics of mung bean based on 10.1016/j.foodchem.2024.138626.

5. L 227; it is better to compare the proximal analysis of gluten-free cocoa cookies with other gluten-free products, not chocolate!

6. L 228; So what is the reason/benefit for adding teosinte flour to produce gluten-free cookies?

7. Elaborate more on parts 3.2 and 3.3. 

8. L 319,320; According to Table 1, teosinte flour had an adverse effect on the nutritional profile of gluten-free cookies.

Author Response

The manuscript evaluates the effects of teosinte flour (Dioon mejiae) on physicochemical characteristics and consumer perceptions of gluten-free cocoa cookies formulated with mung bean (Vigna radiata) flour. The topic is interesting; However, the manuscript has several problems:

  1. Check the spacing between words. For instance, L 42; worldwide; the, or L 43; from the... Check the entire manuscript.

Answer:  Thank you.  We have checked the spacing between words as suggested.

  1. L 51-58 needs some references.

Answer:  As suggested, references have been added.

  1. Please check the reference style. For instance, L 71; "Bastias et al. [6]" not "Bastias et al. (2020) [6]".

Answer: Recommendation was followed. Citations in the text have been checked.

  1. L 80-86; you can elaborate more on the nutritional characteristics of mung bean based on 10.1016/j.foodchem.2024.138626.

Answer:  As suggested, this reference was added.

  1. L 227; it is better to compare the proximal analysis of gluten-free cocoa cookies with other gluten-free products, not chocolate!

Answer: Recommendation was followed.

  1. L 228; So what is the reason/benefit for adding teosinte flour to produce gluten-free cookies?

Answer: Information of the reason/benefit for adding teosinte flour to produce gluten-free cookies was added.

  1. Elaborate more on parts 3.2 and 3.3.

Answer: Done as recommended.

  1. L 319,320; According to Table 1, teosinte flour had an adverse effect on the nutritional profile of gluten-free cookies.

Answer: Sentence was modified.

Reviewer 3 Report

Comments and Suggestions for Authors

This manuscript describes the effect of Teosinte flour addition in mung bean based gluten free cookie formulation on physical properties, sensory analysis as well as consumer perception. The paper is relatively well written and easy to read. However, it needs substantial revision before its acceptance. The following remarks are listed below:

Lines 41 – 42: more complex comparing to what?

Section 2.1: Why did the authors nixtamalized the samples? Please explain.

lines 131-132: Why did the authors decided to replace mung bean flour at 70-100% ratio?

Figure 1: Why are obtained doughs so dark? Mung bean flour is white yellowish in colour. There is 10% of cocoa powder in a recipe. Is this the reason? Still it seems to dark.

Section 2.3. The authors should include also the analysis of used flours.

Lines 155 – 159: What probe was used and what protocol?

The authors could also include some rheological measurement since they provide useful information regarding the changes in dough structure.

Line 171: Why did the authors write CFU/g sample??

lines 175-180 - Please provide the range of ages instead of classification on a boomers, generation X...

line 217 - n=100? - Why question mark?

Line 226-227: Types of what? types of cookies?

lines 231 - 232: how this sentence is relevant for this research? Why did the authors gave comparison to maize?

Table 1. There are no letters of significance for moisture and fat content.

lines 242-243: How is this sentence related to obtained results for hardness

Figure 2. should be placed in 3.4 section, not before.

Section 3.3 and 2.5: From my point of view the microbial analysis are irrelevant for this study and they should be omitted from this manuscript.

Table 2: Why were the hardness values higher for 70% replacement level in comparison to 80%?

Table 2. There are no letters of significance for L , moisture activity and overall liking

Table 2: According to presented data for delta E  values fo 70% and 80% replacement level are not statistically different? Please check

Author Response

This manuscript describes the effect of Teosinte flour addition in mung bean based gluten free cookie formulation on physical properties, sensory analysis as well as consumer perception. The paper is relatively well written and easy to read. However, it needs substantial revision before its acceptance. The following remarks are listed below:

Lines 41 – 42: more complex comparing to what?

Answer:  The sentence was modified to read  “when comparing to products low in carbohydrate content.”

Section 2.1: Why did the authors nixtamalized the samples? Please explain.

Answer: Nixtamalization is a pre-Columbian process that consists of cooking corn grain in an alkaline solution using calcium hydroxide. Consumption of the product without nixtamalization increased the susceptibility of getting pellagra, a disease related to a lack of niacin or Vitamin B3.

lines 131-132: Why did the authors decided to replace mung bean flour at 70-100% ratio?

Answer:  The purpose of the study was to analyze the effect of teosinte flour at a high concentration.  The replacement % was based on our preliminary study.

Figure 1: Why are obtained doughs so dark? Mung bean flour is white yellowish in colour. There is 10% of cocoa powder in a recipe. Is this the reason? Still it seems to dark.

Answer:  Correct--it was the reason.  Black cocoa powder was used and is ultra-Dutch processed, meaning it is treated with an alkaline solution to reduce its acidity. This gives it a smooth texture, dark color, and unsweetened-chocolate highlights.

Section 2.3. The authors should include also the analysis of used flours.

Answer:  The analyses of used flours were added.

Lines 155 – 159: What probe was used and what protocol?

Answer:  The shearing knife blade was used with the protocol of Chakraborty et al. [18]

Chakraborty, S.; Kumbhar, B.; Chakraborty, S.; Yadav, P. process parameter optimization for textural properties of ready-to-eatextruded snack food from millet and legume pieces blends. J. Texture Stud. 2009, 48, 167–174.

The authors could also include some rheological measurement since they provide useful information regarding the changes in dough structure.

Answer:  We did not measure rheological properties.  Thank you for your suggestion.  We will consider it in our future experiment.

Line 171: Why did the authors write CFU/g sample?

Answer:  Modified as noted.

lines 175-180 - Please provide the range of ages instead of classification on a boomers, generation X...

Answer: Recommendation was followed.

line 217 - n=100? - Why question mark?

Answer:  The question mark was deleted.

Line 226-227: Types of what? types of cookies?

Answer: Sentence was modified and corrected.

lines 231 - 232: how this sentence is relevant for this research? Why did the authors gave comparison to maize?

Answer: Maize is one of the closest legumes or cereals that have similar chemical composition to teosinte flour. Maize is also well consumed in the region where teosinte is consumed.

Table 1. There are no letters of significance for moisture and fat content.

Answer: Letters were placed.

lines 242-243: How is this sentence related to obtained results for hardness

Answer: Sentence was modified.

Figure 2. should be placed in 3.4 section, not before.

Answer:  Done as suggested.

Section 3.3 and 2.5: From my point of view the microbial analysis are irrelevant for this study and they should be omitted from this manuscript.

Answer: The samples must be safe prior to the sensory analysis with consumers.  For this reason, we would like to keep these sections.

Table 2: Why were the hardness values higher for 70% replacement level in comparison to 80%?

Answer:  We were not sure as this result was consistent between batches.  It may have been interaction between mung bean and teosinte; however, more studies need to be done to give accurate answers.

Table 2. There are no letters of significance for L , moisture activity and overall liking

Answer: NS was placed as a subscript to indicate that there were no statistical differences between treatments.

Table 2: According to presented data for delta E values for 70% and 80% replacement level are not statistically different? Please check

Answer: Corrections were made, thank you for the observation.